# Hospital Emergency Room Savings via Health Line S24 in Portugal

**Paula Simões** [1,2,*,†] , **Sérgio Gomes** [3] and **Isabel Natário** [1,4,†]

1 Centro de Matemática e Aplicações (CMA), Faculdade de Ciências e Tecnologia, Universidade Nova de Lisboa, 2829-516 Caparica, Portugal; icn@fct.unl.pt
2 Centro de Investigação, Desenvolvimento e Inovação da Academia Militar (CINAMIL), 1169-203 Lisboa, Portugal
3 Direção Geral de Saúde, 1049-005 Lisboa, Portugal; sergiogomes@dgs.pt
4 Departamento de Matemática, Faculdade de Ciências e Tecnologia, Universidade Nova de Lisboa, 2829-516 Caparica, Portugal
* Correspondence: pc.simoes@campus.fct.unl.pt; Tel.: +351-212-948-388
† These authors contributed equally to this work.

**Abstract:** Hospital emergency departments are often overused by patients that do not really need urgent care. These admissions are one of the major factors contributing to hospital costs, which should not be allowed to compromise the response and effectiveness of the National Health Services (SNS). The aim of this study is to perform a detailed spatial health econometrics analysis of the non-urgent emergency situations (classified by Manchester triage) by area, linking them with the efficient use of the national health line, the Saude24 line (S24 line). This is evaluated through the S24 savings calls, using a savings index and its spatial effectiveness in solving the non-urgent emergency situations. A savings call is a call by a user whose initial intention was to go to an urgency department, but who. after calling the S24 line. changed his/her mind. Given the spatial nature of the data, and resorting to INLA in a Bayesian paradigm, the number of non-urgent cases in the Portuguese urgency hospital departments is modeled in an autoregressive way. The spatial structure is accounted for by a set of random effects. The model additionally includes regular covariates and a spatially lagged covariate savings index, related with the S24 savings calls. Therefore, the response in a given area depends not only on the (weighted) values of the response in its neighborhood and of the considered covariates, but also on the (weighted) values of the covariate savings index measured in each neighbor, by means of a Bayesian Poisson spatial Durbin model.

**Keywords:** spatial econometrics; bayesian analysis; autoregressive models; spatio-temporal correlation; poisson; health line; hospital emergency

**JEL Classification:** C49; C11



## 1. Introduction

In Portugal, the clinical priority level is established by the Manchester triage system. This assigns different colors to patients according to their health condition severity, with green, blue and white being the classifications considered to be non-emergency situations. This means that these users could have seen their health situation handled by another more appropriate resource, as for example through a contact with the national health line, the Saude24 line (S24 line). The S24 line was created to improve accessibility to existing services, as well as to rationalize the use of existing resources, by directing users to the most appropriate institutions of the public national health services or giving them advice on self-care. The increased use of hospital emergency rooms by non-urgent patients accounts for very significant efficacy losses of the hospital systems. As a result, the Portuguese hospital urgency service has become one of the major worries of the Portuguese Health Ministry

over recent years. There is daily monitoring and registering of all national emergency services, and this information is made available by the Portuguese Health Directorate-General (DGS) (Daily Monitoring of Emergency Services 2018).

To understand the economic advantages of using the S24 line, from a hospital-management perspective, this work intents to model the expected number of non-urgent emergency cases, by municipality of origin and per year, linking the effective number of registered cases in each hospital emergency department with the performance of the S24 line, through an appropriate savings index. The number of hospital non-urgent cases are then analyzed, considering the hospitals' districts as their influence areas, and these numbers are extrapolated to the municipalities served by the hospitals, within the districts they belong to.

Given the spatial nature of the data, the number of non-urgent cases in the Portuguese urgent hospital departments, by municipality of origin, are modeled considering an autoregressive approach. Regular covariates and a spatially lagged covariate are included by means of a Bayesian Poisson spatial Durbin model. It is plausible to think that the number of non-urgent cases in a given area is related to the number of non-urgent cases in the neighboring areas, driven by effects of the known covariates or others not considered in the modeling Cressie (1993); Lesage and Pace (2009). A possible way of accounting for spatial patterns when modeling a response variable using explanatory variables (covariates) in a generalized linear model is by further including random effects in the predictor, which can be spatially structured. The spatial dependence is included in the model through correlated random effects, where a topology is specified for the spatial units through a spatial weights matrix, $W$, and the spatial structure is considered through a simultaneous autoregressive (SAR) specification. Additionally, because some of the considered covariates can display a spatial dependency, the model may also include spatially lagged covariates. For regular covariates, demographic, socio-economic development and health indicators are considered. The spatially lagged covariate savings index associated with the S24 line is used.

Since one of the covariates of interest is spatially correlated, the most appropriate approach for this type of modeling would be the Durbin spatial model. The spatial modeling formulation proposed here extends the spatial Durbin model for count data, in a Poisson response Bayesian setting, based on the idea that the response in one area depends not only on the (weighted) values of the considered covariates, but also on the (weighted) values of the considered lagged covariates, in this case the lag savings index.

The assessment of the economic impact of the S24 line on the health system, by minimizing hospital emergency care for non-urgent cases, allows us to understand whether the management policies implemented help to reduce the unnecessary costs per capita of health-care, regionally and globally.

This work is organized as follows: first, Section 2 presents an overview of the methodologies used in this work and describes the proposal of a Bayesian spatial Durbin model for Poisson count data; then Section 3 details the data and the considered covariates; Section 4 describes the application of the econometric analysis of the non-urgent emergency situations for assessing the real urgencies in hospitals; finally, Section 5 discusses some of the main conclusions, as well as some possibilities for future work.

## 2. Methods

When modeling dependencies in response across spatial units using spatially correlated effects, the two most common approaches are the SAR (simultaneous autoregressive) model, first presented by Whittle (1954), and the CAR (conditional autoregressive) model, by Besag (1974). These autoregressive specifications are frequently used to model the spatial structure underlying areal data, being known as areal or lattice models. Both models correspond to special cases of describing a general spatial process $\{y_i : i \in S\}$, for which a neighboring structure is defined for the countable set of the area's locations in the indexing set $S$.

Choosing a matrix $W$ for the neighborhood structure, where a topology of the spatial units (or areas) is specified, both models CAR and SAR incorporate spatial dependence into the model covariance structure as a function of $W$ and of a fixed unknown spatial autoregressive parameter.

Consider a vector $y = (y_1, \ldots, y_n)$ of observations on $n$ spatial units and $W$ an $n \times n$ spatial contiguity matrix. A first-order spatial autoregressive model on the response, a SAR model (Arbia 2006; Lesage 1999; Lesage and Pace 2009), is given by

$$\begin{aligned} y &= \rho W y + \epsilon \\ \epsilon &\sim N(0, \sigma^2 I_n) \end{aligned} \tag{1}$$

In this formulation, variation in the response $y$ is solely explained by a linear combination of the response variable in neighboring units, and no other explanatory variables. Parameter $\rho$ is the autoregressive parameter. This model is frequently used for checking the existence of spatial correlation in residuals. The error term $\epsilon$ is assumed to follow a Normal distribution with zero mean and variance-covariance matrix $\sigma^2 I_n$. $\sigma^2$ is a global variance parameter.

An extension of the spatial autoregressive model is known as the spatial lag model (SLM), defined as

$$\begin{aligned} y &= \rho W y + X \beta + \epsilon \\ \epsilon &\sim N(0, \sigma^2 I_n) \end{aligned} \tag{2}$$

where $X$ is a $n \times k$ matrix of explanatory variables and the vector of parameters $\beta$ reflects the influence of these covariates in the $y$ variation. This model combines the standard regression model with a spatially dependent variable model (Goméz-Rubio et al. 2015; Lesage 1999; Lesage and Pace 2009). It can be rewritten so that the response only appears on the left-hand side as

$$\begin{aligned} y &= (I_n - \rho W)^{-1}(X\beta + \epsilon) \Leftrightarrow \\ y &= (I_n - \rho W)^{-1}(X\beta) + \epsilon', \\ \epsilon' &\sim N(0, \Sigma), \end{aligned}$$

with $\Sigma = \sigma^2 (I_n - \rho W)^{-1}((I_n - \rho W)^{-1})^T$ being the variance-covariance matrix defined as a simultaneous autoregressive SAR specification (Lesage 1999; Lesage and Pace 2009; Wall 2004).

The spatial Durbin model adds to a linear model a spatial lag on the dependent variable as well as a spatial lag on the explanatory variables in $X$, being defined as

$$\begin{aligned} y &= \rho W y + X \beta_1 + W X \beta_2 + \epsilon \\ \epsilon &\sim N(0, \sigma^2 I_n) \end{aligned} \tag{3}$$

where $\beta_1$ is the vector of parameters corresponding to the matrix of observed explanatory variables $X$ and $\rho$ represents the spatial lag parameter of the dependent variable. The matrix $(WX)$ represents a spatial lag on the explanatory variables, with associated $k \times 1$ vector of parameters $\beta_2$.

### 2.1. A Bayesian Poisson Spatial Durbin Model

Consider the study region divided into a set of $n$ spatial units, let $y = (y_1, \ldots, y_n)$ represent the number of observed cases of what is being measured in each spatial unit $i = 1, \ldots, n$. Let $e = (e_1, \ldots, e_n)$ be known offsets to be included in the model.

The general log-Poisson regression model is defined as:

$$y_i | \eta_i \sim \text{Poisson}(e_i \theta_i) \tag{4}$$

The counts $y_i$ are assumed to be Poisson distributed with expected value $E(y_i) = \lambda_i = e_i \theta_i$. The parameter of interest is the average relative number of events. It is considered a logarithmic link function, $\log(\theta_i) = \eta_i$, $\theta_i = \exp(\eta_i)$ and $\lambda_i = e_i \exp(\eta_i)$. An offset can be used as a correction factor in the model specification. It represents the denominator of the considered rate, enters the regression on the logarithmic scale and is assumed to have a regression coefficient fixed to 1.

The spatial structure is accounted for through a set of random effects. The model additionally includes regular covariates, $\boldsymbol{X_1}_i = (X_{11i}, \ldots, X_{1ki})$ and a spatially lagged covariates $\boldsymbol{WX_2}_i = (WX_{21i}, \ldots, WX_{2pi})$, measured for each spatial unit $i$. The response in one area depends not only on the (weighted) values of the response in its neighborhood and of the considered covariates, but also on the (weighted) values of the lagged covariates, measured for each neighbor, by means of a Bayesian Poisson spatial Durbin model, defined as

$$
\begin{aligned}
\boldsymbol{y}|\boldsymbol{\eta} &\sim \text{Poisson}(\exp(\boldsymbol{\eta})) \\
\boldsymbol{\eta} &= \rho W \boldsymbol{\eta} + \boldsymbol{X_1} \boldsymbol{\beta_1} + W \boldsymbol{X_2} \boldsymbol{\beta_2} + \log(\boldsymbol{e}) + \boldsymbol{\epsilon} \\
\boldsymbol{\epsilon} &\sim N(0, \sigma^2 I_n)
\end{aligned}
$$

where $\boldsymbol{X_1}$ corresponds to the $n \times k$ matrix of observed explanatory variables with parameters vector $\boldsymbol{\beta_1}$. $W$ is a spatial weight matrix that represents our comprehension of spatial association among spatial units, and $\rho$ represents the spatial lag parameter of the dependent variable. The matrix $(\boldsymbol{WX_2})$ represents a spatial lag on the explanatory variables, with associated $p \times 1$ parameters vector $\boldsymbol{\beta_2}$ and $\boldsymbol{e}$ are known offsets.

## 3. Data

### 3.1. The Non-Urgent Emergency Situations

In Portugal, the clinical priority level is established by the Manchester triage system. This assigns different colors to patients according to their health condition severity, green, blue and white being the classifications considered to be non-emergency situations. The increased demand of these type of patients for emergency hospital services has become one of the most important worries of the Portuguese Health Ministry over recent years.

There is a daily monitoring and registering of all national emergency services and this information is made available by the Portuguese Health Directorate-General (DGS). This work focuses on the number of non-urgent hospital episodes registered in 2016 in Continental Portugal (data from 2013 to 2019 are available). There are 278 municipalities, from which 35 are served by a public hospital with emergency room. The available data on urgencies are solely allocated to the hospital municipality and information on the patient residence is not available. Thus, to get this information desegregated by all municipalities, it is needed to extrapolate the observed number of actually non-urgent episodes to the other municipalities, also served by the hospitals in the larger administrative areas (districts). For this purpose, the number of hospitals and resident population by district are considered (see Figure 1).

Under this framework, it is considered a measure for the number of non-urgent hospital cases, by municipality, considering the district as the influence area of a hospital, given by,

$$
NUH_j = \frac{VAB_j}{(H_j \times RD_j)}, j = 1, \ldots, 18,
$$

where $VAB_j$ is the number of non-urgent episodes, $RD_j$ is the district resident population and $H_j$ is the number of hospitals in district $j$.

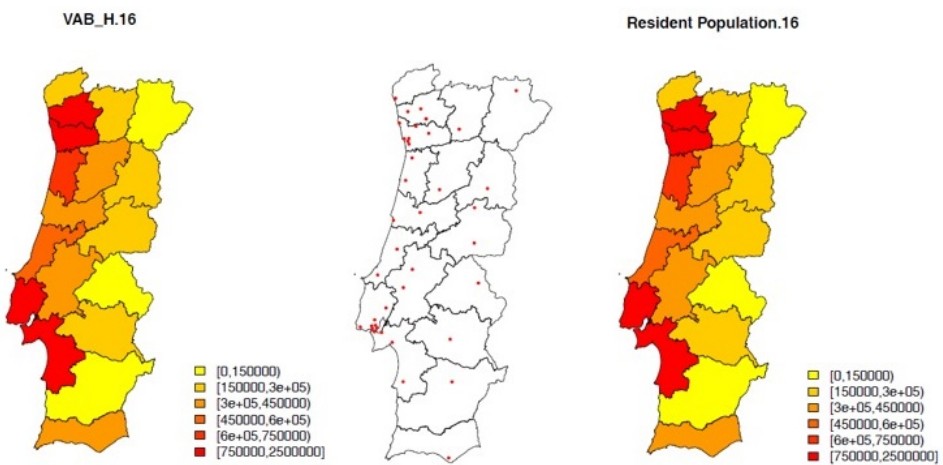

**Figure 1.** Number of non-urgent episodes, locations of the hospitals and resident population, by district, for the 2016 year (**left–right**).

The number of non-urgent hospital episodes (NU) by municipality is then considered to be given as,

$$NU_i = NUH_j \times RM_i, \, i = 1, \ldots, 278, \, j = 1, \ldots, 18$$

where $RM_i$ is the municipality resident population and $NUH_j$ is as described before. $NU$ is represented in Figure 2. This is the response variable to be modeled.

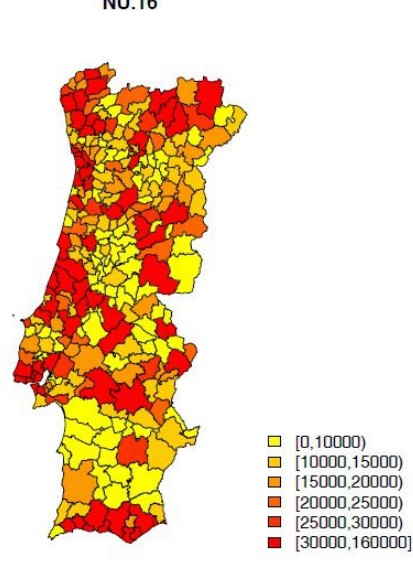

**Figure 2.** Extrapolated number of non-urgent hospital episodes (NU) by municipality for 2016.

The disaggregation of data by municipality was designed considering homogeneity in the use of the available health services within each district.

The model for $NU$ includes some demographic, socio-economic development and health indicators as possible covariates.

From a previous work (Simoes et al. 2019), the average number of years of schooling, the rurality index and the proportion of active population residents stand out as being important for explaining the impact of the S24 line in solving non-urgent emerging situations, mitigating the unnecessary urgent care in hospitals. To include this impact in the current

model, a spatial covariate concerning savings from using the S24 line is derived in the next section.

### 3.2. The Savings Index via Health Line Saúde24

The S24 line offers various services, including Triage, Counselling and Routing (TCR) in disease situations, which represents approximately 90% of the calls to the health line. With the purpose of evaluating the economic impact of the use of the S24 line, concerning the reached savings by avoid unnecessary urgent care in hospitals, the TCR calls whose caller's initial intention was to go to hospital urgency but, according to the final health line disposition, was not necessary, are the relevant ones. These are the calls described as S24 saving calls.

In a previous work, (Simoes et al. 2017), several spatio-temporal Bayesian models for count data were implemented for modeling the S24 saving calls, resorting to INLA methodology, using R-INLA (Bivand et al. 2015; Blangiardo and Cameletti 2015). Spatial structure in data was modeled under both a hierarchical and an autoregressive perspective, concluding that a spatial structure was evident from the analysis. Comparing different model structures, a slight better performance was evident for the autoregressive approach, which is not perhaps the natural choice in terms of the methodology for count data. Several demographic, socio-economic, development and health indicators were investigated as possible covariates for modeling the S24 saving calls counts using a simple log-Poisson model. The expected number of saving calls was included in the model as an offset. Bayesian covariable selection using indicator auxiliary variables under different scenarios (George and McCulloch 1997) was performed. The most significant set of explanatory variables turned out to be the average number of years of schooling, the rurality index and the proportion of active population residents (Simoes et al. 2019).

Following this work, a Savings Index is now defined for each municipality, $i = 1, \ldots, 278$, and for the considered years $t = 2010, \ldots, 2016$,

$$IS_{it} = \frac{(s_{it} p_t - C_{it})}{U_{it}}$$

where $s_{it}$ are the number of savings calls, $p_t$ corresponds to the price of a simple medical appointment in the urgency (which is set every year in a decree-law by the Ministry of Health), $C_{it}$ are the costs associated with annual health line maintenance attributable to municipality $i$ and $U_{it}$ are the total number of calls whose initial intention was go to hospital urgency. *IS* is mapped in Figure 3.

It should be emphasized that this is a simplistic measure which does not consider collateral costs as, for example, those caused by unnecessary urgency increased contagion risk in the population or those caused by increasing the number of patients in the urgency, requiring more human and financial resources.

The savings index detected the municipalities of the metropolitan area of Lisbon as those that most contribute to the economic success of the S24 line. At the same time, it identified the northern interior municipalities as those in which the use of the health line should be encouraged.

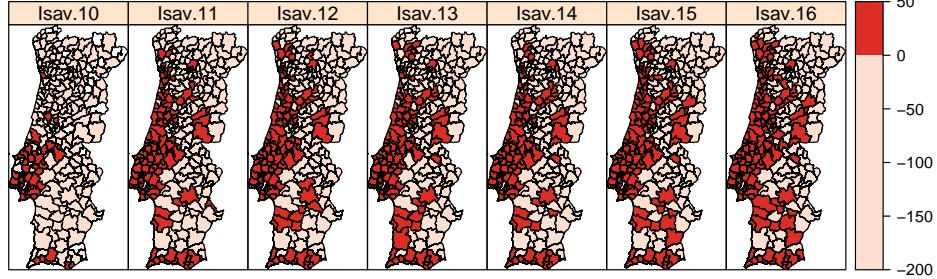

**Figure 3.** The Savings Index from 2010 to 2016.

It was concluded that over recent years, there has been a more comprehensive effectiveness of the S24 line over space in solving the non-urgent emergency situations that could be handled by primary health-care services or in a self-care basis. This suggested that the S24 line is not equally disseminated over the whole territory and, moreover that this is not even associated with the available health services' offer. Consequently, regional directed campaigns of the S24 line use should be considered to be a priority.

Please note that what follows is considered to be the savings index for 2016.

## 4. Results—Modeling Real Urgency Cases in Hospitals

Resorting to INLA methodology (Rue et al. 2009), the Bayesian Poisson spatial Durbin model, presented in Section 2.1, is implemented in R-INLA to model the number of non-urgent hospital cases by municipality in 2016. Different scenarios concerning different sets of regular and spatially lagged covariates were considered. The most significant regular covariates turn out to be the average number of years of **schooling** and the proportion of **active population**; the most significant spatially lagged covariate was the **lag savings index**. Parameter estimates for the selected model are summarized in Table 1. Fitted models were compared by means of their predictive accuracy, using the Deviance Information Criterion (DIC) measure and the Watanabe-Akaike Information Criterion (WAIC) measure (Gelman et al. 2014; Spiegelhalter et al. 2002).

**Table 1.** Parameter estimates (mean, 2.5% and 97.5% quantiles) for the Spatial Durbin Poisson model, for 2016.

| Variable | ID | Coefficients | $(2.5\%, 95\%)$ |
|---|---|---|---|
| Average number of years of schooling | $x_1$ | 0.63 | $(0.50\,; 0.76)$ |
| Proportion of Active population residents | $x_2$ | $-0.24$ | $(-2.80\,; -2.33)$ |
| Rurality index | $x_3$ | 0.15 | $(-0.36\,; 0.65)$ |
| Lag savings index | $x_4$ | $-0.015$ | $(-0.02\,; -0.01)$ |
| Intercept | | 1.72 | $(-0.34\,; 3.81)$ |
| $\sigma^2$ | | 0.48 | $(0.40\,; 0.57)$ |
| $\rho$ | | 0.69 | $(0.63\,; 0.75)$ |

The estimated posterior mean of the spatial main effect and the lag savings index effect are depicted in Figure 4. For the chosen model it stands out that two of the initial regular covariates revealed to be significant: the average number of years of **schooling** per municipality and the proportion of **active population** residents, per municipality; the **spatial component** is quite relevant, which is confirmed by the high value of the estimate of the spatial autocorrelation parameter, significant estimated value of $\rho$ of 0.69; the **spatially lagged** covariate is significant and contributes to a better performance of the selected model, as the model only including regular covariates and spatial random effects did worse.

The considered Bayesian Poisson spatial Durbin model improved the model fit suggesting that this model is more appropriate for these data when compared with the model that do not include the spatially lagged covariate.

The use of the S24 line and its spatial effectiveness (assessed through the savings index behavior) are negatively correlated with the response (number of non-urgent cases in the Portuguese urgency hospital departments) revealing that in a general spatial perspective, the S24 line contributes to avoiding part of the unnecessary urgent care in hospitals. The economic success of the good use of the health line for future assessment of hospital savings is confirmed and it is essential for reducing one of the major factors regarding hospital costs. This model allows description and evaluation of which municipalities the use of the health line should be encouraged, raising awareness so that citizens do not use the hospital urgency unless it is really necessary.

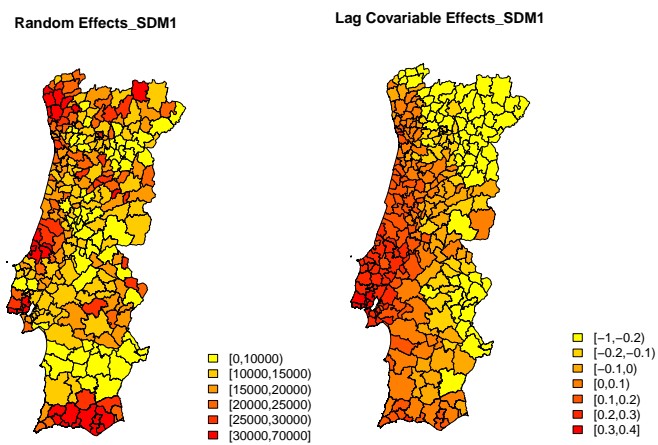

**Figure 4.** Estimated spatial effects for the chosen model for 2016.

## 5. Discussion

This study presents the first steps in the analysis of non-urgent Portuguese emergency cases and their relationship with the S24 savings calls. This was the first time these data on the number of non-urgent cases in Continental Portuguese urgency hospital departments, by municipality of origin, have been modeled. Spatial autocorrelation was taken into account, as observations from geographically close spatial units tend to have more similar values, requiring the selection of a neighborhood structure that specifies the relationships between regions to identify spatial association among them Cressie (1993).

Bayesian Poisson spatial Durbin models were considered for different possible covariate combinations. These models resort to autocorrelated random effects to account for spatial correlation, allowing simultaneously regular covariates and spatial lagged covariates. Additionally, in this quest of considering autoregressive models it was realized that we could go a bit further on the autoregressive modeling, as the spatial Durbin lag model was not yet implemented in a Poisson response Bayesian setting, which was achieved combining a method to fit traditional autoregressive models under the Bayesian INLA approach with a frequentist lag autoregressive Poisson model Bivand et al. (2014); Goméz-Rubio et al. (2015).

Within the aforementioned approaches, the hospital non-urgent emergency situations, for 2016, were studied. The estimation was improved by the addition of spatial structure in the models in an autoregressive way, by regular covariates and by the spatially lagged savings index.

The average number of years of schooling per municipality and the proportion of active population residents per municipality revealed to be significant, standing out as being important in explaining the number of non-urgent cases in the Portuguese hospital urgency departments. The spatial component was quite relevant, which was confirmed by the high significant values of the spatial autocorrelation parameter. The economic success of the good use of the S24 line for future assessment of hospital savings is also confirmed, in terms of the lag savings index behavior, being essential in reducing one of the major factors regarding hospital costs.

In terms of future work, it would be very important to have the residence municipality of the patients so that the extrapolation that is made here would not be necessary. The designed disaggregation of data by municipality, considering that in each district there is homogeneity in the use of that available health services, is a weak point in this modeling. The need to understand the impact of savings index and consequently the use of the national health line on the hospitals' performance, combined with the lack of analysis for this type of problem justifies the taken approach. This assumption has certainly implications for the spatial autocorrelation parameter estimation. However, being the extrapolation based on population density and not on neighborhood criteria, the impact should not be felt on the spatial distribution. It is our view that this strategy for modeling is preferred to any, as

a first approach, to be able to evaluate the hospital-management policies implemented at a national level.

The aim of this project is to develop an economic understanding of the advantages of the health line for the health system and to learn about the political and economic factors that influence health policies at global and regional levels. This enables realization of whether or not these management policies helped to reduce the unnecessary per-capita costs of the health-care, using the savings index and its spatial effectiveness in solving the non-urgent emergency situations. In this context, the analysis should be extended in order to move on to spatio-temporal models in a health econometrics approach (Blangiardo and Cameletti 2015; Cressie and Wikle 2011), develop and implement the temporal and the spatio-temporal effect on the Bayesian spatial Durbin model for count data, what will allow an understanding of one of the major factors regarding hospital costs of Portuguese health-care system, the non-urgent emergency situations, responsible for major efficacy losses of the hospital systems, knowing that with an increase in recent years, presently there are still about 35% of non-urgent cases per year in Portuguese Hospital Care System ( Hughes and McGuire 2003; Sobolev and Levy 2016).

**Author Contributions:** P.S. and I.N. reviewed, discussed, developed the proposed methodologies for the specific data and wrote the paper; P.S. analysed the data and implemented this study in the R software and S.G. collected and made available the S24 line data. All authors have read and agreed to the published version of the manuscript.

**Funding:** This research was funded by national funds through FCT—Foundation for Science and Technology—under the projects UID/MAT/00297/2019 and UID/MAT/00006/2013.

**Institutional Review Board Statement:** Not applicable.

**Informed Consent Statement:** Not applicable.

**Data Availability Statement:** The data presented in this study are available on request from the corresponding author. The data are not publicly available due to privacy.

**Acknowledgments:** This work is financed by national funds through FCT—Foundation for Science and Technology—under the projects UID/MAT/00297/2019 and UID/MAT/00006/2013.

**Conflicts of Interest:** The authors declare no conflict of interest. The founding sponsors had no role in the design of the study; in the collection, analyses, or interpretation of data; in the writing of the manuscript, and in the decision to publish the results.

## Abbreviations

The following abbreviations are used in this manuscript:

| | |
|---|---|
| MDPI | Multidisciplinary Digital Publishing Institute |
| S24 | Portuguese national health line |
| DGS | Portuguese Directorate-General Health |
| SAR | Simultaneous Autoregressive |
| CAR | Conditional autoregressive |
| TCR | Triage, Counselling and Routing |
| INLA | Integrated Nested Laplace Approximations |
| DIC | Deviance information criteria |
| WAIC | Watanabe-Akaike information criteria |

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
