# Peer review of "Hospital Emergency Room Savings via Health Line S24 in Portugal"

_econometrics, doi:10.3390/econometrics9010008_

Round 1
Reviewer 1 Report
Major issues
What does this paper add to the existing knowledge? The method is standard and the application rather scholastic with no attempt to explain any substantive issue. To be acceptable the authors should make a better job in explaining the theoretical underpinnings of their model, the results of their empirical analysis and the policy implications. What does a spatial approach add to the understanding of the phenomenon under study with respect to a more conventional a-spatial analysis? Why should the saving index be negatively spatially correlated? Why should it be correlated with years of schooling? And why negatively with active population? Section 2.0 is unnecessary and can be removed or shrinked in few lines. It is standard literature. The language is a bit cumbersome with many errors. The paper should be thoroughly revised by a native speaker.
MINOR issues
- Language needs a revision. Some sentences have no meaning. For instance:
line 8 « A saving call is a that «
line 201 « the spatially lagged covariate is significant and necessary “. What does it mean necessary ?
line 70 “Choosing a matrix W for the neighbourhood structure, both models CAR and SAR incorporate »
- There are some missing reference such as 83 (Goméz-Rubio 2015);
Reviewer 2 Report
Review of “Hospital Urgency Savings via Health Line S24”
Summary: This paper measures the savings related to the use of a triage system to prevent non-urgent ED visits. The estimates are based on a Bayesian Poisson spatial Durbin model and reveal significant spatial autocorrelation. The results also imply that the triage system is not fully diffused and further gains can be made is use was widespread.
- The number of non-urgent episodes in each municipality is not observed in the data. To overcome this limitation the authors estimate the number of episodes in each municipality by multiplying the non-urgent visits per hospital by the proportion of the municipality population in the district. It seems that this step could induce spatial autocorrelation in that municipalities within a district are highly correlated (they are all proportional to population) whereas the between district variation is actually what is meaningful. I could be misunderstanding the implications of this step. But, even if I am, it should be justified beyond just a lack of data. Is there any value of doing this beyond increasing the number of observations and potentially inducing spatial autocorrelation?
- The savings index measures the savings per urgent call. It is not apparent to me why this is included in the specification. What is the rationale for including the lagged savings index? Presumably the individual making the call is not aware of the savings. A positive value implies that a large number of calls were triaged. Is it that this index measures the general knowledge of the line? Perhaps I am misunderstanding how the health line works. But it is not surprising that an increase in the proportion of “savings calls” reduces the number of non-urgent cases.
- The benefit of the Bayesian Poisson spatial Durbin model over a more rudimentary approach is not self-evident to me. The manuscript could be improved if it described why the this approach is worthwhile. For example, what problems does it overcome?
